# Generative AI for Therapy? Opportunities and Barriers for ChatGPT in Speech-Language Therapy

**Yao Du**
University of Southern California
yaodu@usc.edu

**Felix Juefei-Xu**
New York University
juefei.xu@nyu.edu

## Abstract

Speech-language pathologists (SLPs) are health professionals who work with children and adults with various communication disorders in areas such as speech, language, hearing, and voice. The rise of voice assistants and chatbots brings new opportunities for SLPs and points to new opportunities and barriers when adopted during clinical service delivery. This paper explores the potential adoption of ChatGPT in speech-language therapy for individuals with receptive and expressive language disorders. By offering SLP critique for ChatGPT's responses to multiple therapeutic use cases, limitations and solutions for improving generative AI tools for speech-language therapy are also discussed.

## 1 Introduction and Background

The American Speech-Language-Hearing Association (ASHA) defines communication disorder as "an impairment in the ability to receive, send, process, and comprehend concepts or verbal, nonverbal and graphic symbol systems" (ASHA, 2023) which impacts individuals' ability to communicate at the speech level (*e.g.*, difficulty producing speech sounds), language level (*e.g.*, difficulty formulating sentences and narrative stories), cognitive level (*e.g.*, difficulty retrieving and remembering salient information), and/or perceptual level (*e.g.*, difficulty processing and hearing). Speech-language pathologists (SLPs) are trained professionals who screen, assess, evaluate, treat, and monitor pediatric, adult, and geriatric clients with communication disorders across various educational and medical settings (*e.g.*, schools, hospitals, and rehabilitation centers). Despite the increased adoption of web and mobile technology tools during therapy among SLPs (Du et al., 2022), there is a growing need and knowledge gap regarding how to apply artificial intelligence (*e.g.*, natural language processing tools) to improve the efficiency and effectiveness of clinical service delivery across nine areas of SLP's professional scope of practice, also referred to as "ASHA Big 9" (ASHA, 2023). Prior research by (Wang et al., 2021) has investigated the utility of using OpenAI GPT models to analyze conversations during therapy sessions between caregivers of individuals with dementia and their therapists; however, this prior work was conducted during problem-solving therapy rather than speech-language therapy under the ASHA Big 9 domains. The present paper examines the implications of using ChatGPT for one specific ASHA Big 9 domain - language. Language disorders can impact different age groups of individuals such as children with autism (Tager-Flusberg & Caronna, 2007) and adults with aphasia (Theodoros et al., 2008), further hindering their academic learning, social communication, and activities of daily living.

ChatGPT (OpenAI, 2022), based on InstructGPT (Ouyang et al., 2022), is the latest manifestation of large language models (LLMs) with a user-friendly conversational interface. InstructGPT is an enhanced version of GPT-3 (Brown et al., 2020), achieving better outputs with 100x fewer parameters (1.3B vs. 175B). The biggest algorithmic advancement in InstructGPT is the use of reinforcement learning from human feedback (RLHF) with two sets of special-purpose datasets: (1) the demonstration dataset for training supervised policy in the initial model where a prompt is sampled from the prompt dataset and human labeler demonstrates the desired output behavior; (2) the comparison dataset for fine-tuning the previously trained model using RL, where a model-written message is randomly selected and several completions are sampled, and then a human AI trainer is asked to rank them. Using these reward models, the model can be fine-tuned using proximal policy

optimization (PPO) methods iteratively. The dataset mainly contains text prompts submitted to the OpenAI API. To train the first InstructGPT models, human labelers were asked to write prompts themselves as an initial source of instruction-like prompts are needed to bootstrap the process.

## 2   OPPORTUNITIES AND BARRIERS USING CHATGPT FOR TREATING LANGUAGE DISORDERS

SLPs' language intervention typically focuses on either expressive language (*e.g.*, using words in sentences to express ideas and thoughts) or receptive language (*e.g.*, understanding the ideas and thoughts expressed in words) using various therapy materials, such as using printed pictures or words on flashcards to target language production at the word/phrase/sentence/conversation level, and using toys and games to target various communication functions (*e.g.*, questioning, answering, commenting) (Leochico et al., 2022). To explore ChatGPT's potential for clinical utility, an ASHA-certified speech-language pathologist evaluated responses of ChatGPT for proposed language intervention activities for one of the ASHA Big 9 domains, receptive and expressive language. Table 1 shows the analysis of ChatGPT's response for 8 proposed language intervention activities. Based on the 8 suggested therapy use cases proposed by ChatGPT, the evaluating SLP offered critiques related to adopting ChatGPT as 1) a valuable assistive therapy tool for generating therapy materials, 2) a chatbot that simulates human-like communication during therapy, and 3) an intelligent assistant for the therapist. Specifically, ChatGPT has the potential to become an innovative tool in SLPs' clinical service delivery through translating and generating therapy materials (vocabulary development, narrative skills, language comprehension, literacy, bilingual therapy) and simulating human-like communication (grammar and syntax, pragmatics, cultural competence). These functions can offer clients, such as those with autism, a low-pressure communication environment without directly communicating with humans (Benford & Standen, 2009), improve clients' ability to use additional novel therapy materials for practice outside therapy (Leochico et al., 2022), and reduce clinicians' workload and therapy preparation costs (Cirrin et al., 2003). A detailed report of an example use case which used ChatGPT to generate targeted word lists was documented by a research assistant of the authors, demonstrating the feasibility of the aforementioned implementation opportunities with a 14-year client with cerebral palsy (Price, 2023).

One unique opportunity for using ChatGPT for speech-language therapy is that depending on the client population and needs, it is possible that using existing database or knowledge base may already be adequate for clinicians to adopt generative AI tools such as ChatGPT because many conventional speech-language therapy materials are typically drawn from functional activities of daily living which are already available in the public domain. Additionally, ChatGPT could be used with other generative AI tools (Table 2) such as DALL-E to replace conventional "worksheets and flashcards" so that clients can participate in therapy using only nonfictional content but also fictional content to work with diverse and novel therapy tasks and activities. Table 2 shows the type/modality of generative AI and clinical implications for speech-language therapy. Future clinicians can envision a multi-modality generative AI which can serve as an AI conversational partner to directly communicate with clients, leading to a fundamental change in human-AI collaborative therapy across multiple ASHA Big 9 domains.

Despite these opportunities, a preliminary critique of ChatGPT's responses revealed additional barriers to clinical implementation. First, using open-source training data leads to questionable data accuracy. For example, the common issue in deep learning models leads to low-data regime issues, which can impose bias, especially in low incidents and rare disease conditions. As a result, therapy materials generated may not be as tailored and customized to individuals with a less frequent therapy task compared to more generative activities that are more readily available on the internet. Second, using ChatGPT also raised additional questions regarding data ownership and usage. If ChatGPT requires client data input, how to determine which part of the patient-generated language data are protected sensitive information and who has the initial and final access and ownership of the private information and content? Discussions among multiple stakeholders (*e.g.*, clients, clinicians, caregivers, and data scientists) regarding data governance are required. Third, this paper did not perform a critical analysis of the implications of ChatGPT on social communication. Conversational AI tools are prone to inappropriate social language, hate speech, and emotionally disturbing conversations (The Verge, 2023), which requires collaborative research effort across domains such as AI safety, fairness, *etc*.

URM STATEMENT

The authors acknowledge that at least one key author of this work meets the URM criteria of ICLR 2023 Tiny Papers Track.

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

## A    APPENDIX

Table 1: Analysis of ChatGPT's Response for Intervention in "Language", One of the ASHA Big 9 Domains.

| ChatGPT's Response | SLP Critique for Use Cases |
|---|---|
| Vocabulary Development: ChatGPT can be used to help individuals build their vocabulary. Clients can ask ChatGPT to define new words they come across or ask for synonyms for words they already know. | Word definitions and synonyms are common therapy targets for children with expressive language disorders. Conventional therapy focuses on activities such as using flashcards, worksheets, and dictionary with support from SLPs. ChatGPT could be utilized to **generate therapy materials**. |
| Grammar and Syntax: ChatGPT can be used to help individuals practice correct grammar and syntax. Clients can ask ChatGPT to provide feedback on their sentence structure and ask for suggestions on how to rephrase their sentences. | Asking for and providing feedback on rephrasing and using correct sentence structure are activities provided by SLPs. ChatGPT could be utilized as **a simulated human-like therapy assistant** as SLPs deliver therapy activities. |
| Narrative Skills: ChatGPT can be used to help individuals practice their storytelling skills. Clients can ask ChatGPT to tell a story, and then practice retelling the story in their own words. | During conventional intervention, SLPs may work with clients using clients' personal experiences, commercially available storybooks, and/or create novel stories for therapy. ChatGPT could be utilized to **generate therapy materials**. |
| Pragmatics: ChatGPT can be used to help individuals practice their social communication skills. Clients can engage in conversation with ChatGPT and practice using appropriate nonverbal cues, turn-taking skills, and topic maintenance. | During conventional therapy, clients need to practice conversations with human beings; with ChatGPT, clients can engage in conversations (currently via text chat only) with ChatGPT which can work as **a simulated human-like therapy assistant**. |
| Comprehension: ChatGPT can be used to help individuals practice their comprehension skills. Clients can ask ChatGPT questions about a topic they are learning and practice answering comprehension questions. | During conventional therapy, comprehension activities are typically provided by clinicians using materials such as worksheets and stories; ChatGPT could **generate therapy materials** such as sample learning content and facilitate comprehension questions. |
| Literacy: ChatGPT can be used to help individuals develop their literacy skills. Clients can engage in conversation with ChatGPT about a topic they are learning and practice reading and writing skills. | Conventional literacy intervention typically involves SLPs working with SLPs using a variety of reading and writing aids (*e.g.*, graphic organizers) to target specific reading and writing goals. ChatGPT could be utilized to **generate therapy materials** such as creating templates to support reading and writing. |
| Bilingualism: ChatGPT can be used to help individuals who are bilingual practice their language skills in both languages. Clients can engage in conversation with ChatGPT in one language and then switch to another language to practice their language skills in both languages. | Conventional bilingual therapy typically requires support from an interpreter. With ChatGPT, **translation of therapy materials** can be readily available for clinicians. It is important to note that localization and translation may not always be culturally and linguistically sensitive to clients' needs. |
| Cultural Competence: ChatGPT can be used to help individuals develop their cultural competence by providing information about different cultures and their languages. Clients can engage in conversation with ChatGPT about cultural differences in language usage and practice using culturally appropriate language. | In conventional therapy, clinicians need to rely on searching information prior to therapy to increase cultural competence of diverse cultures and languages. ChatGPT could serve as **a simulated therapy assistant for information seeking**, although the reliability of sources can be questionable. |

Table 2: Type/Modality of Generative AI and Clinical Implications for Speech-Language Therapy.

| Type | Description | Clinical Use Cases |
|---|---|---|
| Text Generation: ChatGPT (OpenAI, 2022), LLaMA (Touvron et al., 2023), Bard (Google, 2023), *etc*. | The SOTA text generation is primarily based on transformer-based large language models that are trained on large-scale language corpus using self-supervised learning. The user will provide a text-based prompt, and the text generation model will produce a plausible answer. The conversation can continue alternately between the user and the chatbot, and the generative model can produce answers based on the chat history so far. | (1) Generate therapy stimuli for use during therapy and after therapy for additional exercises; (2) Provide or brainstorm therapy activity ideas and generate lesson plans for therapy; (3) Serve as a chatbot to communicate with clients directly. |
| Image Generation: DALL-E 2 (OpenAI, 2023), Midjourney Midjourney (2023), Stable Diffusion (StabilityAI, 2023), Imagen (Saharia et al., 2022), *etc*. | The SOTA image generation is primarily based on vision transformer or CNN-based probabilistic diffusion models. The user will provide source images and accompanying text prompts as instructions, and the image generation model will produce a plausible picture according to the user's specifications in the instruction. The user can also specify the exact bounding box locations or object mask regions in order to precisely generate certain contents at certain locations. | Generate visual stimuli for therapy materials (*e.g.*, picture cards, visual schedules, stories, memory aids) and parent education materials (*e.g.*, additional homework exercises) |
| Video generation: Make-a-Video (Singer et al., 2022), Imagen Video (Ho et al., 2022), *etc*. | The SOTA video generation is primarily based on extending the image generation capabilities with additional temporal consistency constraints. The user will provide source image, or image set, or video clips and accompanying text prompts as instructions, and the video generation model will produce a plausible video clip according to the user's specification in the instruction. | Create additional homework videos with avatars for caregiver training |
| Music generation: MusicLM (Agostinelli et al., 2023), *etc*. | The SOTA music generation is primarily based on transformer or RNN-based large-scale sequence-to-sequence models. The user will provide a source melody and accompanying text prompts as instructions, and the music generation model will produce the music clip according to the user's specification in the instruction. | Support therapy materials for language, memory, and voice (*e.g.*, melodic intonation therapy, reminiscence therapy, singing exercises) |
| Multi-modality generation AI: putting everything together | The aforementioned generative capabilities can be combined either sequentially or in parallel to produce multi-modality and multimedia contents. Also, cross-modality multimedia generation will be made possible such as text2 (video+music), music2text, music2video, image2video, image2music, image2(text+video), *etc*. | Create simulated robots as future human-AI cooperative therapy delivery agents |

