# OpenReview forum: "Generative AI for Therapy? Opportunities and Barriers for ChatGPT in Speech-Language Therapy"
_ICLR.cc/2023/TinyPapers — Submitted to Tiny Papers @ ICLR 2023_

### Official Review · Reviewer_rg3D · 2023-03-29

**Confidence:** 2

**Summary Of Contributions:**

The paper discusses potential use cases for a chat bot like ChatGPT to be used by speech-language pathologists (SLPs). Several potential uses are enumerated, and both the benefits and limitations are explored

**Rating:**

Great Start (GS): a submission which meets some of the reviewing criteria but has room for improvement

**Strengths And Weaknesses:**

Strengths:

-	The paper clearly indicates that there is potentially immense value for SLPs to use chatbots such as ChatGPT.

Weaknesses:

-	The paper reads more like a research proposal than a research paper.

-	The contributions from the paper are not clear. Most of the body of the paper is a literature review, and it is not clear what additional content was provided by the authors.

-	Much of contributions appear to be found in the appendix. For example, while there is a brief discussion about the limitations of ChatGPT, much of the more substantive critique can be found in the appendix.

-	Most of the ChatGPT prompts focus on asking ChatGPT about the field itself rather than a demonstration of the various proposed uses in the paper. It is not clear why ChatGPT’s perception of its implications is relevant to the discussion.


**Suggested Changes:**

-	Clearly indicate the contributions from the paper.

-	Exploring ChatGPT’s own perception of its implications focuses on its use for information recall as opposed to specific clinical tasks. Greater focus on demonstrating ChatGPT in a clinical setting, presumably a simulated one, would significantly enhance the paper.

-	For Tables 1-3, the inconsistencies and errors are not obvious to non-SLPs. More clarity would be necessary for a lay person.

---

> ### Author Response · Authors · 2023-06-01
> **Response to the reviewer**
>
> Response to the reviewer: We appreciate the suggested changes to improve this paper. We have revised our writing with 1) reordered paragraphs, and 2) improved clarity through direct explanation of the most relevant tables (Tables 4 & 5) in the main text and removal of unrelated tables (Tables 1-3).

---

### Comment · Area_Chair_UJgf · 2023-06-08
**Archival Criterion Check**

This work meets the threshold for archival, contains the URM statement, and is deanonymized.

---

### Meta-Review · Area_Chair_UJgf · 2023-04-08

**Recommendation:** Invite to revise
**Confidence:** 2

**Metareview:**

The reviewer points out that the paper can viewed more as a research proposal (or an Opinion piece in my view). I agree with this. However, I do not see this as disqualifying for Tiny Papers per se, although I would encourage the authors to make this a bit more clear in the Abstract and the Introduction.

I also agree with the reviewer's last Weakness point, that the prompts are focused on ChatGPT's view on "the field itself rather than a demonstration of the various proposed uses." The paper confuses application of LLMs (how the can be _applied_ in a therapeutic setting) with the evaluation of their generative abilities (how good are they at generating convincing new text in response to a prompt, which is dissociable from therapy). Thus, I do encourage the authors to replace or augment these prompts with better demonstrations of the use cases.

For example, the "SLP Critique" column of Table 4 can be integrated in the main text as examples of how ChatGPT can be used by SLPs. The authors should illustrate ways of facilitating this by providing instructions or guidelines on how to get ChatGPT to display the kind of behavior desirable for such applications.

**Summary:**

The paper discusses the idea that ChatGPT or other LLMs could be used for therapy. There are no clear points about how this can be done technically (e.g., prompt engineering to get ChatGPT to generate the write type of text) or practically (how useful this actually would be in the real-world).

**Reason For Not Giving A Higher Recommendation:**

The idea that ChatGPT can be used many different settings is a widely discussed one. In that regard, this paper does not add much to the discussion. It is a great start for the particular use case of Speech Language Therapy, but unfortunately there is not enough relevant materials for this purpose.

**Reason For Not Giving A Lower Recommendation:**

N/A

---

> ### Author Response · Authors · 2023-06-01
> **Response to the MetaReviewer**
>
> Response to the Metareviewer: We thank the reviewer for pointing out several issues for this paper, which have been addressed correspondingly in the updated PDF, namely:
> 1) Weak abstract and introduction: improved with shorter descriptions.
> 2) More clarity on use cases: integrated Table 4 (Table 1 in the updated PDF) SLP critique in the main text as examples of “use cases of ChatGPT by SLPs” followed by additional generative AI tools and use cases for speech-language therapy (Table 5, which is Table 2 in the updated PDF), along with additional blog article demonstrating a detailed case study in the reference.

---

### Decision · Program_Chairs · 2023-04-09

Revision accepted; invite to archive

---

> ### Author Response · Authors · 2023-05-31
> **Opt in for archival**
>
> All authors would like to opt in for archival.